# The Fabrication of Portland Composite Cement Based on Pozzolan Napa Soil

**DOI:** 10.3390/ma14133638

**Published:** 2021-06-29

**Authors:** Mawardi Mawardi, Illyas Md Isa, Alizar Ulianas, Edtri Sintiara, Fadhlurrahman Mawardi, Rizky Zalmi Putra

**Affiliations:** 1Department of Chemistry, Faculty of Mathematic and Natural Sciences, Universitas Negeri Padang, Padang 25173, Indonesia; alizarulianas@fmipa.unp.ac.id (A.U.); edtri.sintiara@gmail.com (E.S.); rizkyzalmi.putra@yahoo.com (R.Z.P.); 2Department of Chemistry, Fakulti Sains dan Matematik, Universiti Pendidikan Sultan Idris, Tanjong Malim 35900, Perak, Malaysia; 3Department of Chemistry, Universitas Andalas, Padang 25175, Indonesia; fadhelmawardi@gmail.com

**Keywords:** cement, pozzolans, Napa soil, composite materials, compressive strength

## Abstract

The objective of this study is to investigate Napa soil’s potential as an alternative additive in producing Portland composite cement. The Napa soil of Tanah Datar district, West Sumatra, Indonesia is a natural material which contains SiO_2_ and Al_2_O_3_ as its major components. The parameters used were the fineness of the cement particles, the amount left on a 45 μm sieve, the setting time, normal consistency, loss on ignition, insoluble parts, compressive strength and chemical composition. The composition of Napa soils (% *w*/*w*) used as variables include 4, 8, 12 and 16%. Furthermore, 8% pozzolan was used as a control in this research. The results showed that the compressive strength of Napa soil cement which contained 4% Napa soil was much better compared to that of the control on the 7th and 20th day. Furthermore, all the analyzed Napa soil cements met the standard of cement as stipulated in Indonesian National Standard, SNI 7064, 2016.

## 1. Introduction

Cement, as the binding material for stones and concretes, plays an important role in every construction activity; therefore, one cannot separate cement and construction [1,2,3]. In 1824, Joseph Aspidin from England carried out the calcination of the mixture of limestone and clay. Through the developments of science and technology, research on high-quality cement has been carried out to reduce the disadvantages of ordinary cement [4].

Portland composite cement (PCC) is a new variant type of commercial cement that is similar in characteristics with Portland cement in general, but has better quality, is ecofriendly and cheap [3]. The composition of the cement materials includes clinker, gypsum and additive materials such as fly ash and trace or pozzolan [3,4]. Therefore, the Indonesian cement industry uses pozzolan as an additive material.

Pozzolan is a material rich in aluminum and silicon [5], which in its finely divided form and in the presence of water, chemically reacts with calcium hydroxide. This reaction gives rise to hydrated calcium silicate and other byproducts [4,6]. Although pozzolan does not bind materials as effectively as cement, the water it contains causes it to form binding substances through certain reactions [7]. Therefore, pozzolan may be substituted with another alternative material with high silica oxide content to produce cement [8]. Unfortunately, the high abundance of Napa soil in West Sumatera has not been fully utilized (Figure 1a). In West Sumatera, an estimates 150,000,000 tons (154 Ha) of Napa soils can be found in various areas such as Pesisir Selatan Regency, Tanah Datar Regency, Solok Regency, and Lima Puluh Kota Regency (Figure 1b).

Napa soil is produced naturally, originating from the weathering of feldspathic rocks and carried out by endogenous forces that do not move from the parent rock or origin. As this soil does not move, it has a purer nature, in the sense that it is not mixed with other types of soil and material. Napa soil contains alumina silicate in the ratio of SiO_2_/Al_2_O_3_, which is approximately 1.25:3.43. In addition, it contains SiO_2_, Al_2_O_3_, Fe_2_O_3_, TiO_2_, CaO, K_2_O and Na_2_O, with percentages of 70.43, 20.52, 3.67, 0.40, 2.70, 1.26% and trace, respectively [9]. Meanwhile, Napa soil has not been effectively used to its full potential. Therefore, in this research, Napa soil was used to produce a type of PCC that varies in additive materials such as clinker, gypsum and limestone [10], where it was used as a natural pozzolan. Based on its composition, Napa soil was categorized as natural pozzolan to produce PCC.

The main compounds in cement include: elite (3CaO·SiO_2_ (C_3_S); belite 2CaO·SiO_2_ (C_2_S); aluminate, 3CaO·Al_2_O_3_ (C_3_A); and ferrite, 4CaO·Al_2_O_3_·Fe_2_O_3_ (C_4_AF). When cement and water react together, the product formed is called calcium–silicate–hydrated (C-S-H) gel, and it is the main agent that binds the cement and particles together within the concrete. The following chemical equations illustrate the reactions between C_2_S and C_3_S [11,12].
2(3CaO·SiO_2_) + 6H_2_O **→** 3CaO·2SiO_2_·3H_2_O + 3Ca (OH)_2_(1)
2(2CaO·SiO_2_) + 4H_2_O **→** 3CaO·2SiO_2_·3H_2_O + Ca (OH)_2_(2)

## 2. Materials and Methods

### 2.1. Tools and Materials

The tools used in this research were crusher ME-100 Jaw Crusher 5”X8”, mini mill, mixer and its dish, stirrer Toni TechnikToniMI, mixing knife, Vicat tools, ebonite ring and glass plate, Zwick Roell Toni SET, alpine air jet 200LS-N, cube mold, pressing equipment, oven Carbolite CWF 1300, Blaine tool Toni Technik Toni Trol. Glass equipment, X-ray fluorescence (XRF) model Epsilon3 PANanalytical Netherlands, X-ray diffraction (XRD) model XPERT PRO PANanalytical Netherlands, and thermogravimetric analyzer (TGA) model DTG 60 Shimadzu, Kyoto, Japan.

The materials used were Napa soil from subdistrict of Lintau of Tanah Datar district, clinker, gypsum, limestone, water, Herzog pill, NaOH, HCl (Merck, Jakarta, Indonesia, pa grade), MM indicator and Ottawa sand.

### 2.2. Preparation of Napa Soil

Lumps of Napa soil were crushed using crusher. Then, its fine particles were dried under sunlight in order to evaporate its water content and then sieved to pass through a 90 µm sieve.

### 2.3. Preparation of Composite Cement

Cement samples were prepared by thoroughly mixing various ratios of clinker, gypsum, limestone and pozzolan or Napa soil bases. In order to determine the characteristics of the cement produced, the analysis of its particles’ fineness, loss of ignition, insoluble residue, particles left on 45 µm sieve, normal consistency, setting time and the compressive strength of the mortar produced was carried out [13]. Meanwhile, XRF (X-ray Fluorescence) was used to determine its chemical composition. 

### 2.4. Characterization of Composite Cement

#### 2.4.1. XRF Analysis

About 30 g of Napa soil cement and Napa soil were added using 2 pellets of Herzog pill, and ground. The fine sample produced was added into the XRF analyzer to carry out the characterization step.

#### 2.4.2. Blaine Analysis

Determining the cement’s fineness was carried out using Blaine analysis. About 2.8948 g sample was placed into the Blaine apparatus for the characterization process.

#### 2.4.3. Lost on Ignition (LOI) Analysis

Analysis of LOI was carried out according to the loss of water and CO_2_ when the cement was ignited. Firstly, 1.000 g of cement was ignited at the temperature of 1000 °C for 15 min. Then, it was placed in a desiccator for the cooling process.

#### 2.4.4. Insoluble Residue Analysis

One gram of the sample was placed into 250 mL beaker glass, followed by the addition of 25 mL of HCl and aquadest dropwise until it reached 100 mL. The solution was boiled, filtered and rinsed using aquadest. Then, the precipitate formed was placed in a beaker which contained 1% of 100 mL NaOH. Additionally, two drops of methyl red indicator and a few drops of HCl were added until the color of the solution become red. It was then filtered and rinsed with hot water and NH_4_NO_3_ solution. Finally, the solid was placed in a furnace at 1000 °C for 30 min and weighed. The percentage of insoluble part was calculated using this formula:% insoluble part = (mass of precipitate/mass of sample) × 100%(3)

#### 2.4.5. Sieving Analysis

A 20 g of sample was placed on 45 µm sieve and analyzed using sieving machine for 3 min. After that, the remaining samples left in the sieve were weighed. The percentage of sieving analysis was calculated using this formula:% sieving analysis = (mass after sieving/mass before sieving) × 100%(4)

#### 2.4.6. Normal Consistency Analysis

A 650 g of sample was placed in the mixing dish followed by the addition of 156 mL of water. Then, the cement was added to the water and mixed for 30 s and left to set for 15 s. The paste that set on the bowl was collected and further mixed for another 1 min. After that, the paste produced was collected again, molded into a ball shape and pushed through the Vicat ring.

#### 2.4.7. Setting Time Analysis

Water was added to 650 g of cement to produce paste according to normal consistency analysis. Then, the penetrating test was carried out using Vicat pins of 1 diameter every 10 min. The first setting was marked when the pin penetrated more than 25 mm and stopped when the Vicat pin could not penetrate further, which implies that it has reached the final setting.

#### 2.4.8. Compressive Strength Analysis

Compressive strength analysis was carried out based on the standard procedure of PT. Semen Padang, which refers to the Indonesian National Standard (SNI 2049:2015). The samples used for this analysis were 500 g of cement, 1375 g of standard sand and 276 mL of water. Firstly, the cement was placed in the dish and water was added. Using a mixer, the samples were mixed with a low velocity (140 ± 5) gradient per minute for 30 s and the sand was slowly added to the mixture with low velocity. Meanwhile, the velocity was increased to a gradient of (285 ± 10) per minute for 30 s. The mixer was stopped and left for 1.5 min, and within the first 15 s the dough attached to the wall of the bowl was lowered and the stirring bowl was closed with a lid to prevent evaporation of water in the paste. Then, the mixer was turned on for 60 s at medium velocity, then switched off, and subsequently turned on again for another 15 s. After pouring the mixture into each cube, they were each slowly pressed 32 times at ±4 mm cube. All residual mixtures on the cube mold were scraped with a knife, while the surface was softened with a spoon. After molding, the sample was placed in a humid environment for one whole day, and its surface was covered to prevent contact with water. Subsequently, the sample was removed from the mold, soaked in water containing calcium oxide, and then placed in an antirust humid room until it was analyzed. Evaluation of its compressive strength was performed on day 3, 7, and 28. Compression of the sample was carried out using tools that were calibrated.

## 3. Results and Discussion

The use of Napa soil from the subdistrict of Lintau of Tanah Datar district as an alternative substitution material for pozzolan was carried out in different variations, while the pozzolan was used as the control. The addition of Napa soil is the source of SiO_2_ that reacts with the byproduct of the reaction between 3CaO·SiO_2_, 2CaO·SiO_2_ and water (Equations (2) and (3)). 3CaO·SiO_2_·3H_2_O is the important factor that contributes to the strength of the cement; the more 3CaO·SiO_2_·3H_2_O formed, the greater the strength of the mortar to withstand a load [12]. Then, the Ca(OH)_2_, which is the byproduct of the reaction between C_3_S and C_2_S, reacts with SiO_2_ in the pozzolan as shown in the equation below:3Ca(OH)_2_ + SiO_2_ + H_2_O → 3CaO·SiO_2_·3H_2_O,(5)

The reform of 3CaO·SiO_2_·3H_2_O increases the compressive strength of the mortar. Ca(OH)_2_ should be minimized as a byproduct because it fills a bigger space in the structure that causes breakages in the binding of the hardening cement paste and cracks in the cementation [14,15]. The variation in the amount of the Napa soil (% *w*/*w*), as additive, in producing composite cement is shown in Table 1.

### 3.1. XRF Analysis of Composite Cement

The chemical composition of composite cement using XRF is shown in Table 2. The table shows that the quantities of SiO_2_, Al_2_O_3_, and Fe_2_O_3_ and MgO are proportional to the quantity of Napa soil in composite cement, which is due to SiO_2_ being the main composite of the Napa soil. Meanwhile, the composition of CaO is inversely proportional to the quantity of Napa soil, which is good, as the source of CaO is limestone, which is important to balance the proportion of Napa soil. In addition, MgO is one of the compounds that affect the quality of the cement and it is important to make sure that the level of MgO in the cement is less than 5.0% [16]. This is because the presence of high amounts of Mg(OH)_2_ and periclase can weaken the cement used in construction.

After the determination of the percentages of SiO_2_, Al_2_O_3_, Fe_2_O_3_, CaO, MgO, and SO_3_, the compositions of C_3_S (3CaO·SiO_2_), C_2_S (2CaO·SiO_2_), C_3_A (3CaO·Al_2_O_3_) and C_4_AF (4CaO·Al_2_O_3_·Fe_2_O_3_) were calculated using the formula stipulated in Indonesian National Standard (SNI 2049:2015) [17]:
C_3_S = (4.071 × %CaO) − (7.600 × %SiO_2_) − (6.718 × %Al_2_O_3_) − (1.430 × %Fe_2_O_3_) − (2.852 × %SO_3_)C_2_S = (2.867 × %SiO_2_) − (0.7544 × C_3_S)C_3_A = (2.650 × %Al_2_O_3_) − (1.692 × %Fe_2_O_3_)C_4_AF = (3.043 × %Fe_2_O_3_)

The composition of C_3_S, C_2_S, C_3_A, and C_4_AF of cement with variation of Napa soil shown in Table 3. C_3_S contributes significantly to the strength in the beginning. Meanwhile, C_2_S also contributes to the cement strength in the longer term. C_3_A affects the compressive strength after 28 days and becomes less effective after one or two years. C_4_AF does not affect the compressive strength significantly [18].

### 3.2. Analysis of the Fineness

The fineness analysis of the composite cement particle was carried out using a Blaine instrument as stipulated in Indonesian National Standard of SNI 2049:2015. The bar chart that shows the relationship between the fineness of the cement particle and the percentage of Napa soil composition is shown in Figure 2. The fineness of the cement particle and the surface area of the cement are proportional to the amount of Napa soil due to its softness and brittleness. The composite cement produced by using Napa soil as an additive material has smooth and fine granules, so the cement tends to form agglomerates when analyzed using SEM (Figure 3) [19].

The quantity of Napa soil used in producing composite cement is proportional to its grinding time until it reaches 3800–4200 cm^2^/g. In Figure 4, CNS4 gives a good result as it requires only a short setting time and high compressive strength, probably due to its fine cement particles [13].

### 3.3. Analysis of the Loss on Ignition

The value of the loss of ignition (LOI) should be minimal because it is proportional to the level of mineral that can be parsed in the cement. The effect of using Napa soil on the loss of ignition (LOI) is shown in Figure 5. Composite cement with 16% Napa soil shows a low LOI value, which means the amount of the mineral parsed is low, which is due to low quantity of limestone in composite cement from all cement samples. The percentage of limestone in cement is inversely proportional to the quantity of Napa soil in cement. From the results, it is clear that mortar with high LOI, containing ground bagasse ash, has slower comprehensive strength development when compared to those with low LOI. However, both of them displayed similar compressive strength over time [20].

### 3.4. Analysis of Insoluble Part of Cement

The effect of Napa soil usage on the percentage of the insoluble part is shown in Figure 6. It was discovered that the high percentage of Napa soil added enhanced the percentage of insoluble part of cement, which is due to high content of silica in Napa soil. The identification of the insoluble part was carried out by dissolving cement as shown in the reaction below:

When cement reacts with hydrochloric acid, HCl, it forms a water-soluble salt compound, such as CaCl, AlCl_3_, and FeCl_3_, H_2_SiO_2_ and H_4_SiO_4_, in the form of a gel that is difficult to dissolve. If cement reacts with NaOH, it will form a compound in the form of salt, as shown in the reaction:
H_2_SiO_2_ + NaOH **→** Na_2_SiO_4_ (Dissolved)H_4_SiO_4_ + NaOH → Na_4_SiO_4_ (Undissolved)Na_4_SiO_4_ + HCl → H_4_SiO_4_ + NaCl

Meanwhile, H_4_SiO_4_ is calcinated at 1000 °C for 1 h to form
H_4_SiO_4_ → H_2_O + SiO_2_

SiO_2_ cannot be dissolved in water and acid solvents. The insoluble part of the cement should be minimal as possible and SiO_2_ needs to completely react. CNS1 shows the low percentage of insoluble parts. CNS2 that has the same composition, with a lower insoluble part percentage than the control. This shows that the cement containing Napa soil has lower percentage of insoluble parts than the cement containing pozzolan. Portland cement contains pozzolan, a non-cementing substance called insoluble residue, which influences properties of the cement such as compressive strength [21]. Therefore, in order to control the amount of non-cementing materials within the cement, ASTM standards prevent the quantity of insoluble residue from exceeding 0.75%. This is lower than the British standard of 1.5%. However, the normal consistency and setting times are unaffected by the addition of insoluble residue to the cement. Despite all these data, this hypothesis needs to be further researched in order to be considered completely reliable [22].

### 3.5. Analysis of the Rest on Sieve of 45 µm

Figure 7 shows the analysis of Napa soil usage on a 45 µm sieve. It shows that the quantity of Napa soil used in producing the cement is inversely proportional to the percentage of the 45 µm sieve. The relationship chart between the use of Napa soil to the residue above the 45 µm sieve shows that the relative increase in the amount of Napa soil used does not affect the amount of cement that escapes from the 45 µm sieve, with an average of 17.215%. The particle size of cement is one of the important factors that influences the speed of its reaction with water, and also influences the compressive strength of the cement. If the cement has a fine particle size, when making mortar, it covers the cavities in the mortar mixture. Therefore, the product is denser and has high compressive strength. Of all cement samples, the control had the lowest number of particles above the sieves. It can be understood that many control cements pass through the 45 µm sieve, which speeds up the binding time and increases the compressive strength of the cement. Arvaniti et al. [23] discovered that the fly ash which was tested met the STM C618 [24] fineness criterion, with not less than 34% of the substance remaining on the 45 µm sieve. Furthermore, 20% of the slag from the blast furnace met the ASTM C989 [25] fineness criterion.

### 3.6. Analysis of Normal Consistency

The effect of the of Napa soil usage and its normal consistency is shown in Figure 8. It shows that composite cement that contains 16% Napa soil (CNS4) has higher values of normal consistency than cement CNS1, CNS2, and CNS3, which is due to the size of the cement particles as well as low surface area that reduces the water distribution and eventually requires a greater volume of water to be of normal consistency.

### 3.7. Analysis the Setting Time

Generally, the cement hydration process coupled with the hydration degree makes a key contribution to properties of hardened cement [26]. The content of C_3_A, particle size and gypsum in a cement will significantly affect its setting time. The effect of C_3_A, which reacts with water, on the setting time is shown in the reaction below:3CaO·Al_2_O_3_ + 6H_2_O → 3CaO·Al_2_O_3_·6H_2_O 3CaO·Al_2_O_3_ reacts with gypsum to form ettringite, which covers the surface of the 3CaO, while Al_2_O_3_·6H_2_O and 3CaO·Al_2_O_3_ block the hydration reaction of 3CaO·Al_2_O_3_ and prevent the hardening process. Increasing amount of pozzolanic materials can increase the hydration process, hence increasing the required time period [27]. This influences the amount of time the material takes to set [15].

From Figure 9, it can be deduced that the setting time is proportional to the amount of Napa soil in the produced cement. When a large amount of Napa soil added to the cement, this produces a larger particle size, which leads to a longer setting time. Furthermore, the SNI standard starting setting time is 45 min. The maximal final setting time is 375 min. It means that all samples used meet the national standard for cement. The correlation between the percentage of Napa soil and the setting time of the cement is shown in Figure 9.

### 3.8. Analysis of Napa Soil Usage to the Compressive Strength

Napa soil’s effect on compressive strength is displayed in Figure 10, which shows a decrease with increasing composition of the Napa soil as the additive contributing silica–alumina. The cements CNS2 and PCC have the same composition, but the control contains pozzolan, while CNS2 contains Napa soil. Meanwhile, at days 7 and 28, the compressive strength of the mortar of the cement containing 4% Napa soil (CNS1) was higher than that of the pozzolan.

The amount remaining on the 45 µm sieve is another factor that influenced the compressive strength; the less the material that rested on the sieve, the greater the amount of cement that has a particle size smaller than 45 µm. Therefore, the mortar formed more solidly and the small particles of the cement filled its pores. The quantity of Cement CNS4 left on the sieve was 15.93%.

XRD analysis was performed to determine the mineral content of composite samples such as calcium silicate hydrate (C-S-H), calcium aluminosilicate hydrate (C-A-S-H), portlandite (Ca(OH)_2_), and ettringite minerals in the sample (Figure 11). The intensity of calcium hydroxide (Ca(OH)_2_) 2θ = 20.71 and ettringite 2θ = 22.9 increased with the increasing percentage in the sample. The peak of calcium silicate hydrate (at 2θ = 26.55) increased with the percentage in the sample. An increase in intensity also occurred at 2θ = 29.41, which indicates a very fast hydration process in the sample. The higher intensity of calcium hydroxide (Ca(OH)_2_) was able to form C-S-H and increase the strength of the sample.

The effect of increasing temperature on the mass loss (weight loss) of the sample and determining the temperature of the decomposition process of the cement sample using TGA analysis is shown in Figure 12.

The TGA analysis shows a significant weight loss for all of the cement samples. The first weight loss, at 30 °C to 100 °C, occurred due to the evaporation of water from the pores on the surface of samples. Second, the weight loss shown at 100 °C to 450 °C occurred due to dehydration of calcium silicate hydrate and C-A-S-H. The drastic change at 475 °C to 750 °C related to the decomposition of Ca(OH)_2_ during the hydration process, as shown in equation below:Ca(OH)_2_**→** CaO + H_2_O (evaporation)(6)

Decarbonation process of CaCO_3_ at 800 °C:Ca(OH)_2_ + CO_2_ **→** CaCO_3_ + H_2_O (carbonation)(7)
CaCO_3_ **→** CaO + CO_2_ (decarbonation)(8)

Then, the percentage of calcium hydroxide and water content was calculated as:Mass of calcium hydroxide (%) = [(W_400_ − W_580_)/W_580_] × (74/18) × 100(9a)
where the molecular weights of calcium hydroxide and water are 74 and 1, respectively.
Water content (%) = [(W_80_ − W_580_)/W_580_] × 100(9b)

Figure 13 shows the calculated percentage of calcium hydroxide in the samples. Sample CNS12% (CNS3) contains the highest percentage of calcium hydroxide, while the water content is 1.529%. The lowest percentage of calcium hydroxide was found in sample CNS8% (CNS2), which was caused by the formation of CSH gel in the sample.

## 4. Conclusions

All analyzed samples met Indonesia National Standard of SNI 2049:2015. The use of Napa soil as pozzolan in producing cement showed good result percentages compared to control. Meanwhile, the parameters used were the loss of ignition, the amount remaining on 45 µm sieve, normal consistency, fineness of cement particle, insoluble part, setting time, compressive strength and chemical composition. From this study, the results led to the conclusions below:(a)Composite cement with 16% Napa soil shows a low LOI value, which means that the mineral parsed is low and this is due to low quantity of limestone in composite cement from all cement samples.(b)The increase in the amount of Napa soil used does not affect the relative amount of cement that escapes the 45 µm sieve, with an average of 17.215%.(c)The compressive strength of the mortar on days 7 and 28 of the cement containing 4% Napa soil was higher compared to that of pozzolan. Furthermore, the XRF analysis of the cement showed that the most common chemical compounds produced in the cement were SiO_2_ and Al_2_O_3_.

## Figures and Tables

**Figure 1 materials-14-03638-f001:**
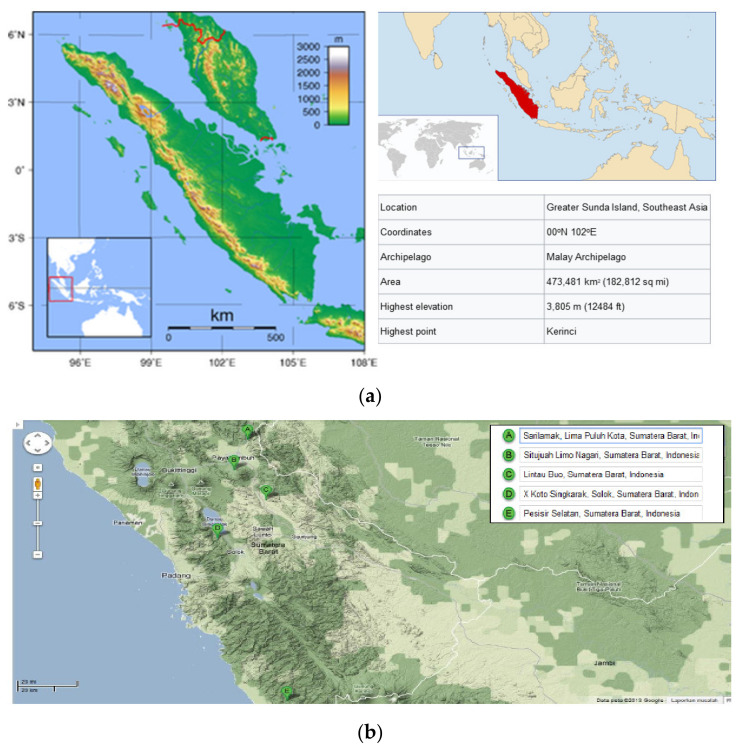
(**a**) Napa soil distribution in Sumatera Island. Topography of West Sumatera, (**b**) and map of West Sumatera province which contains the largest amount of Napa soil [9].

**Figure 2 materials-14-03638-f002:**
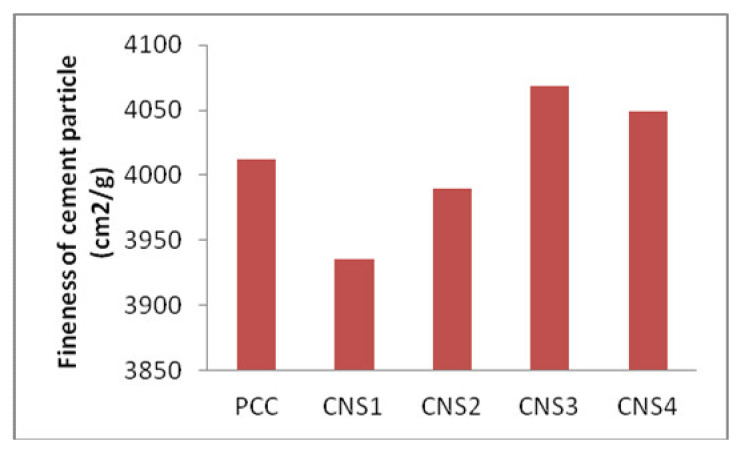
The relationship between the fineness of the cement particle and the percentage of Napa soil composition.

**Figure 3 materials-14-03638-f003:**
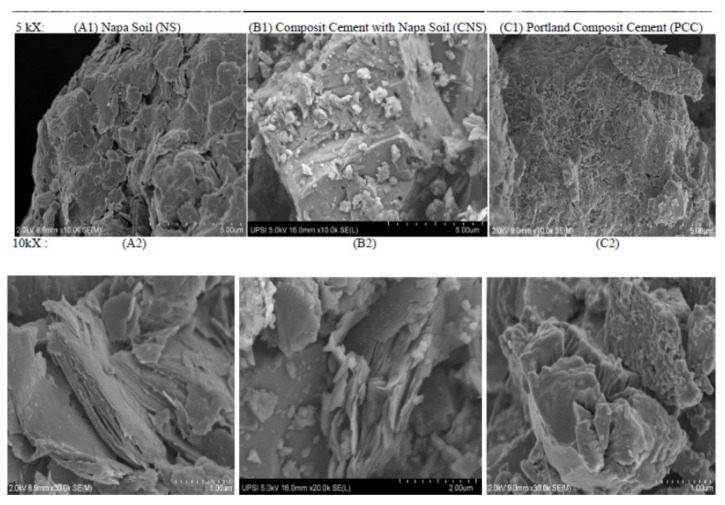
SEM image (**A**) Napa soil (NS), (**B**) CompositecCement with Napa soil (CNS) and (**C**) Portland composite cement (PCC).

**Figure 4 materials-14-03638-f004:**
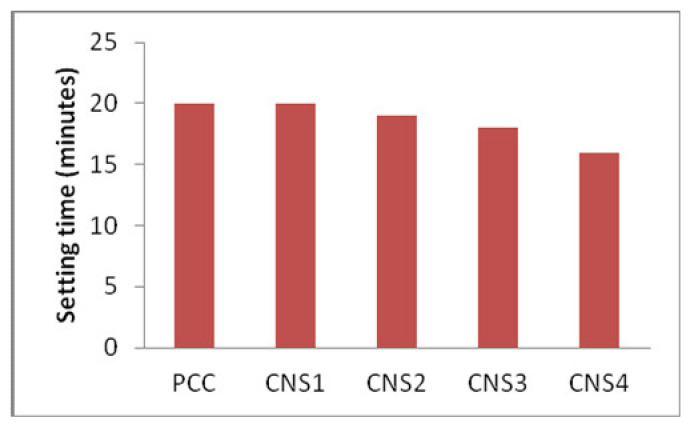
Analysis of Napa soil usage to the setting time (minutes) with its variation.

**Figure 5 materials-14-03638-f005:**
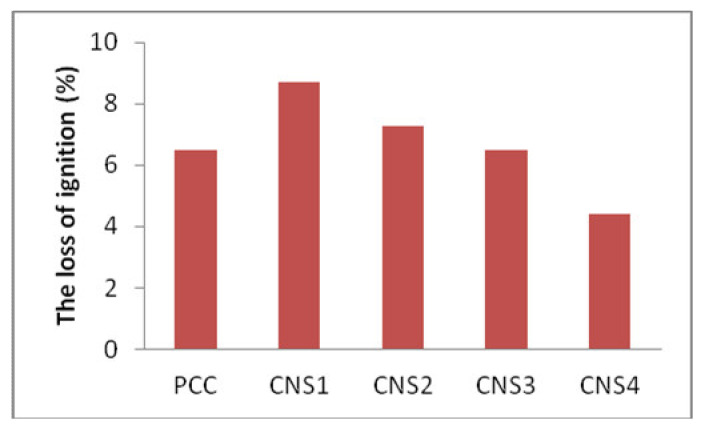
The effect of Napa soil usage on the loss of ignition (LoI).

**Figure 6 materials-14-03638-f006:**
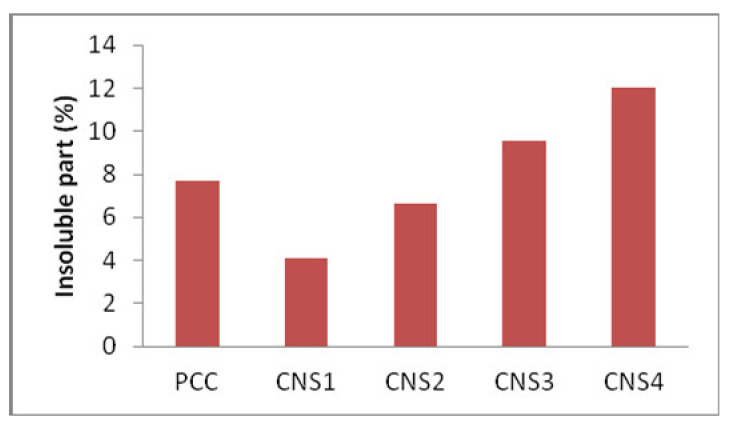
The effect of Napa soil usage on the insoluble part.

**Figure 7 materials-14-03638-f007:**
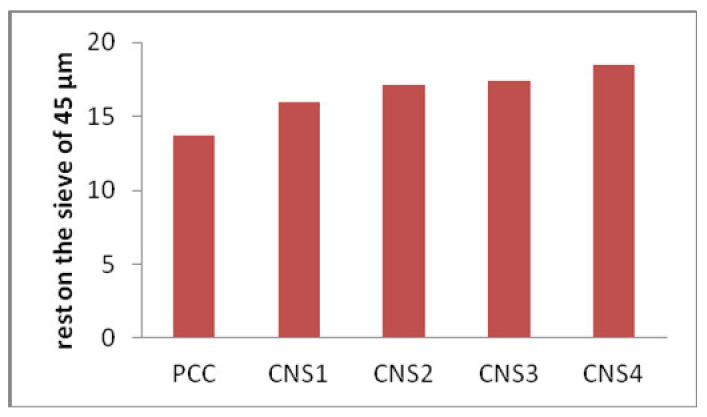
The effect of Napa soil usage on the percentage remaining on 45 µm sieve.

**Figure 8 materials-14-03638-f008:**
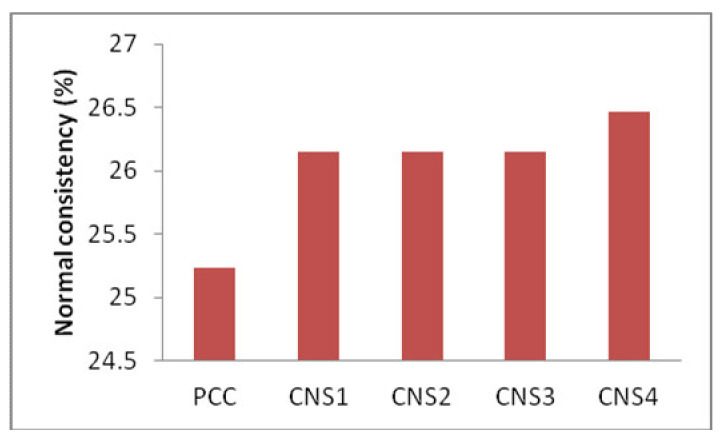
The effect of the utilizing of Napa soil to normal consistency.

**Figure 9 materials-14-03638-f009:**
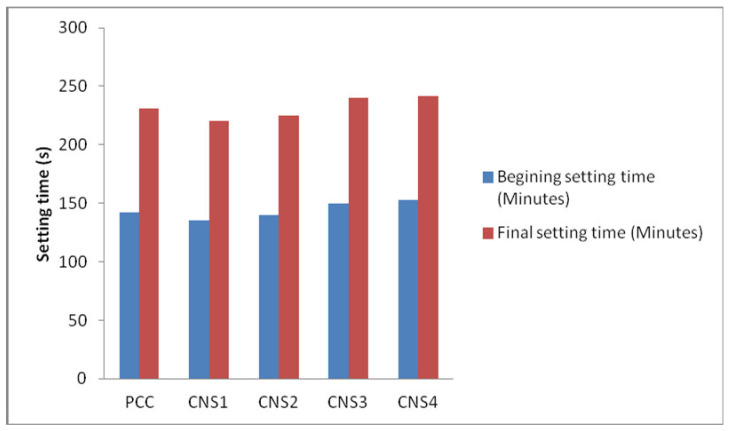
The relationship between the percentage of Napa soil, starting setting time, and final setting time of the cement.

**Figure 10 materials-14-03638-f010:**
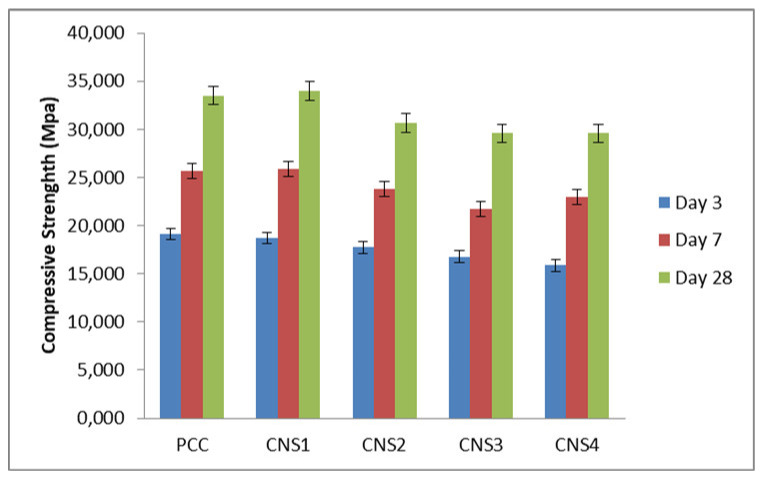
The relationship between the percentage of Napa soil (*n* = 5) and compressive strength of the cement.

**Figure 11 materials-14-03638-f011:**
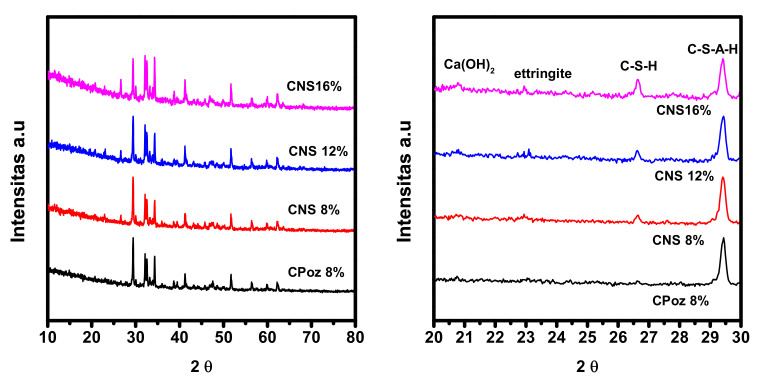
XRD analysis of portland composite cement based on pozzolan Napa soil on day 28.

**Figure 12 materials-14-03638-f012:**
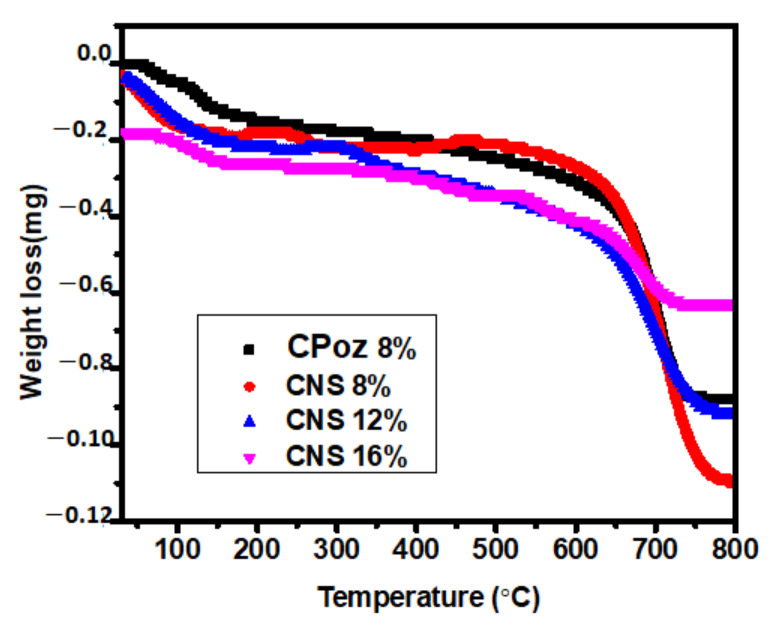
TGA analysis of portland composite cement based on pozzolan Napa soil on day 28.

**Figure 13 materials-14-03638-f013:**
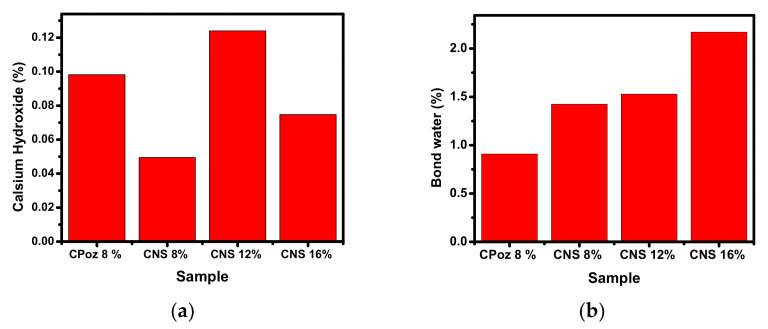
(**a**) Mass of calcium hydroxide, (**b**) Mass of bound water.

**Table 1 materials-14-03638-t001:** The variation of materials in composite cement.

Sample	Clinker (%)	Gypsum (%)	Limestone (%)	Pozzolan (%)	Napa Soil (%)
PCC	76%	4%	12%	8%	-
CNS1	76%	4%	16%	-	4%
CNS2	76%	4%	12%	-	8%
CNS3	76%	4%	8%	-	12%
CNS4	76%	4%	4%	-	16%

**Table 2 materials-14-03638-t002:** Chemical composition (% *w/w*) of cement with variation of Napa soil.

Sample	SiO_2_	Al_2_O_3_	Fe_2_O_3_	CaO	MgO	SO_3_
PCC	22.00	6.10	3.29	59.36	0.91	1.79
CNS1	19.98	5.49	3.16	61.96	0.96	1.84
CNS2	21.27	6.04	3.37	60.04	1.00	1.75
CNS3	22.79	6.63	3.64	57.89	1.08	1.77
CNS4	24.72	7.34	3.92	55.49	1.11	1.67

**Table 3 materials-14-03638-t003:** The composition of C_3_S, C_2_S, C_3_A, and C_4_AF of cement with variation of Napa Soil.

Sample	C_3_S	C_2_S	C_3_A	C_4_AF
PCC	23.67	45.22	10.60	10.01
CNS1	53.74	16.74	9.20	9.62
CNS2	32.39	36.55	10.30	10.25
CNS3	7.67	59.55	11.41	11.08
CNS4	−21.63	87.20	12.82	11.93

## Data Availability

The data presented in this study are available on request from the corresponding author.

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
