# Peer review of "The Fabrication of Portland Composite Cement Based on Pozzolan Napa Soil"

_materials, 2021, doi:10.3390/ma14133638_

Round 1

Reviewer 1 Report

Dear Authors,

Congratulations for your work on the investigation of napa soil's potential as an alternative additive in producing Portland composite cement.

I consider that your paper addresses na interesting topic in new materials, therefore, is suitable for this Journal.

In general, this paper is clear, the topic is contextualized, and data collection strategy and treatment are well conducted. The parameters used (the fineness of cement particles, the rest on 45 μm sieve, setting time, normal consistency, loss on ignition, insoluble parts, compressive strength and chemical composition) were appropriated and sufficient compositions of Napa soil were considered. Control was assured and experiments were well conducted according to standards. All the references are pertinent.

However, Authors were negligent on the presentation of units, tables, and figures.

I strongly recommend Authors to review, as for example

gram and g in lines 80, 88, 100, 106, 112

ml and mL in lines 92

min. and minutes inline 89, 97, 100 and 109

In addition, Authors should check references 23 and 27 that are missing in the manuscript, as well as Table 4 (line 282).

Figure 3 and 4 should be mentioned in the manuscript and in line 294, figure 11 must be replaced by 11.

To sum up, I think the manuscript is a valuable case-study and fits the standards of the journal but must be slightly improved in order to be published.

Author Response

Dear Editor and Reviewers,

Thank you for your feedback on my manuscript entitled “Production of Portland Composite Cement Based on Pozzolan Napa Soil”. After taking into consideration of the comments and suggestions, I have made the necessary changes and correction as required. I wish to resubmit my revised manuscript for your kind consideration for publication. I have also attached herewith my responses to the comments for your information.

Yours faithfully,

(Illyas Md Isa)

Responses to Comments

#Reviewer1

C1

:

However, Authors were negligent on the presentation of units, tables, and figures. I strongly recommend Authors to review, as for example: gram and g in lines 80, 88, 100, 106, 112; ml and mL in lines 92; min and minutes inline 89, 97, 100 and 109. In addition, Authors should check references 23 and 27 that are missing in the manuscript, as well as Table 4 (line 282). Figure 3 and 4 should be mentioned in the manuscript and in line 294, figure 11 must be replaced by 11.

Response:

Thank you for the suggestion. The presentation of units, tables, and figures has been corrected based on the suggestions given by the reviewer. The references 23 and 27 have been added to the manuscript and the Table 4 has been changed to Figure 9.  Figure 3 and 4 have been mentioned in the manuscript and Figure 11 have been replaced by 10 in line 294.

Reviewer 2 Report

The paper is interesting, and the authors performed a detailed investigation, especially from a chemical point of view. In the opinion of the reviewer, some modifications have to be made in the parts concerning the mechanical performances:

  1. In the section 2.4.8, are mortar samples made in accordance with some standard? The composition seems to be that of EN 196-1 (2005), but the geometry of the samples and tests setup are not indicated.
  2. The sentence written in the lines 294-295 is not clear: the strength of all the samples increases (and not decreases) with time.
  3. In the diagram of Fig.10, strength has to be measured in MPa. How many specimens have been tested for each test? Please, report the scatter of test result.
  4. If C3S is decreasing with the content of Napa soil (as shown in Fig.11), it could be interesting to observe what happens to compressive strength after 28 days. Probably the strength increases and reaches that of reference cement.
  5. Finally, the conclusion have to be rewritten by reporting a bulleted list of the main results

Author Response

Dear Editor and Reviewers,

Thank you for your feedback on my manuscript entitled “Production of Portland Composite Cement Based on Pozzolan Napa Soil”. After taking into consideration of the comments and suggestions, I have made the necessary changes and correction as required. I wish to resubmit my revised manuscript for your kind consideration for publication. I have also attached herewith my responses to the comments for your information.

Yours faithfully,

(Illyas Md Isa)

Responses to Comments

#Reviewer2

C1

:

In the section 2.4.8, are mortar samples made in accordance with some standard? The composition seems to be that of EN 196-1 (2005), but the geometry of the samples and tests setup are not indicated.

Response:

Thank you for a good comment. Mortal samples made based on the standard procedur of PT. Semen Padang which refers to the Indonesian National Standard (SNI 2049:2015). We have already added in text.

C2

:

The sentence written in the lines 294-295 is not clear: the strength of all the samples increases (and not decreases) with time.

Response:

Thank you for the suggestion. The compressive strength of one sample increased from day 3, 7, and 28. However, the compressive strength decreased with the increase of Napa soil in the cement. The sentence has been corrected in the text (section 3.7).

C3

In the diagram of Fig.10, strength has to be measured in MPa. How many specimens have been tested for each test? Please, report the scatter of test result.

Response:

We agree and appreciate the good comments. The unit of compressive strength in Figure 10 have been coverted to MPa as suggested by the reviewer. There are five samples have been analysis, one of them contains pozzolan and being control cement, while the other four contains Napa soil with varying composition. The results of the compressive strength for each sample are shown in Figure 10 (with scatter of test result).

C4

If C3S is decreasing with the content of Napa soil (as shown in Fig.11), it could be interesting to observe what happens to compressive strength after 28 days. Probably the strength increases and reaches that of reference cement.

Response:

Thank you for the suggestion. Figure 11 has been changed to Table 3 based on data of Table 2 (section 3.1). C3S decreased with the content of Napa Soil. This is because the composition of CaO decreased with the content of Napa soil, while other chemical compositions such as SiO2, Al2O3, and Fe2O3 increased with the content of Napa Soil (Table 2). Based on Table 3, we can see that CNS1 has the highest percentage of C3S. Meanwhile, CNS4 has the lowest percentage of C3S. This is corresponds to the compressive strength data at day 28, that cement CNS1 has the highest compressive strength, while cement CNS4 has the lowest compressive strength, which means C3S contributes significantly to the compressive strength.

C5

The conclusion have to be rewritten by reporting a bulleted list of the main results.

Response:

Thank you for the suggestion. The conclusions have been written by a bulleted list as suggested by reviewer (section 4).

Reviewer 3 Report

Suggested comments and corrections for the Authors of the paper are following:

The title:

Generally the title refers to the content of the article and analyzed problems.

However there is no information about production process, its technology. It was mentioned only about the production of the modified cement. Analyzes (study) of its properties are presented mostly.  

Section 3

Line 140: In the beginning of the text, to check numbers of equations in brackets if they are proper.

Section 3.1

In Table 2, there is no composition of referenced PCC as in Table 1 is presented and in all Figures.

Below the Table 2 it is written that “…the composition of C3S, C2S, C3A and C4AF were calculated using the formula …” however only these formula are presented but results of these calculations are not shown.

Values of compositions for each cement sample could be shown e.g. in the table or in the graph.

Section 3.2

Lines 182, 183: Wrong comment to the Fig.2 that “the curve that shows the correlation …”. The relationship is possible for function y(x). Correlation is a statistical number in case of determination of regression function and its agreement to the y(x) function of experimental values, which belong to the same file.

Line 184:  Improper number of Figure 1 (Fig.2, should be)

Line 188:  Improper number of Figure 2 (Fig.3, should be)

The Fig.2 is composed of values of different cement samples, with pozzolana and with a Napa soil. Only values of Napa soil can be linked with determination of regression function and correlation. In this Figure, five value dots should be separated and only compared. Better type for presentation can be the bar graph type.

Except above there is no information if values are single measurements or average and of what many measurements were made. In case of average values information about the standard deviations should be given as well.

The same problem is with others Figures.

The description of the Fig.2 should be: “The relationship …”

Line 197:  Improper number of Fig. 3 (Fig.4, should be)

Figure 4 can not be with linked value dots, but separated because there are no values on the horizontal axis and they are for different types of cements. Better to use the bar graph type.

Section 3.3 – Section 3.7

Figures 5 – 11 can not be with linked value dots, but separated because there are no values on the horizontal axis and they are for different types of cements. Better to use the bar graph type.

For CPoz8% there are two colors of dots – black and red in Figures 7 – 8 without any description at Figures and explanation in the text.

Section 3.6

Line 282:  To check the written number of Table 4 because there are only two tables in whole article.

Line 287:  Term “correlation” in Figure 9 is improper according to comments for Figure 2. It can be determined as relationship. Better to use double bar graph type.

Section 3.7

Line 294:  Number of the Figure should be 10.

Comments to the Fig.10 and Fig.11  the same as to the Fig.2.

To check the description of the vertical axis of the Fig.11

Line 320:  It is written that “All cements have a high Tricalcum Silicate (C3S) content” does not agree with the Fig.11 because the values of the graph of C3S are different, higher and lower.

To check values scale on the vertical axis because some of them are negative.

Is it possible for C3S  to have values below 0 at Fig.11 ?

Author Response

Dear Editor and Reviewers,

Thank you for your feedback on my manuscript entitled “Production of Portland Composite Cement Based on Pozzolan Napa Soil”. After taking into consideration of the comments and suggestions, I have made the necessary changes and correction as required. I wish to resubmit my revised manuscript for your kind consideration for publication. I have also attached herewith my responses to the comments for your information.

Yours faithfully,

(Illyas Md Isa)

Responses to Comments

#Reviewer3

C1

Generally the title refers to the content of the article and analyzed problems.

However there is no information about production process, its technology. It was mentioned only about the production of the modified cement. Analyzes (study) of its properties are presented mostly.  

Response:

Thank you for the best suggestion. We have already changed the production to fabrication.

C2

Line 140: In the beginning of the text, to check numbers of equations in brackets if they are proper.

Response:

Thank you for the suggestion. We already checked and numbers of equations in brackets.

C3

In Table 2, there is no composition of referenced PCC as in Table 1 is presented and in all Figures.

Response:

We are grateful to you for your detail observation. We already corrected for all figures and tables.

C4

Below the Table 2 it is written that “…the composition of C3S, C2S, C3A and C4AF were calculated using the formula …” however only these formula are presented but results of these calculations are not shown.

Values of compositions for each cement sample could be shown e.g. in the table or in the graph.

Response:

Thank you for a good comment. We already corrected and added the composition of C3S, C2S, C3A, and C4AF on Table 3.

C5

Lines 182, 183: Wrong comment to the Fig.2 that “the curve that shows the correlation …”. The relationship is possible for function y(x). Correlation is a statistical number in case of determination of regression function and its agreement to the y(x) function of experimental values, which belong to the same file.

Response:

Thank you for the suggestion. We have been changed the ‘correlation’ into ‘relationship’ on the text (section 3.2)

C6

Line 184:  Improper number of Figure 1 (Fig.2, should be)

Line 188:  Improper number of Figure 2 (Fig.3, should be)

Response:

We are grateful to you for your detail observation. We have already corrected on the text as the suggestion of the reviewer.

C7

The Fig.2 is composed of values of different cement samples, with pozzolana and with a Napa soil. Only values of Napa soil can be linked with determination of regression function and correlation. In this Figure, five value dots should be separated and only compared. Better type for presentation can be the bar graph type.

Response:

Thank you for the suggestion. We have already corrected on the text and changed to bar chart.

C8

The description of the Fig.2 should be: “The relationship …”

Response:

Thank you for the suggestion. We have been corrected on the text as the suggestion of the reviewer.

C9

Figure 4 can not be with linked value dots, but separated because there are no values on the horizontal axis and they are for different types of cements. Better to use the bar graph type.

Response

Thank you for the suggestion. We have already corrected on the text and changed to bar chart.

C10

Figures 5 – 11 can not be with linked value dots, but separated because there are no values on the horizontal axis and they are for different types of cements. Better to use the bar graph type.

Response :

Thank you for the suggestion. We have already corrected on the text and changed to bar chart.

C11

For CPoz8% there are two colors of dots – black and red in Figures 7 – 8 without any description at Figures and explanation in the text.

Response:

Thank you for the suggestion. We have already corrected in the text.

C12

Line 282:  To check the written number of Table 4 because there are only two tables in whole article.

Response:

We apologize for the mistake. Actually we mean is Figure 9. We have already corrected in the text (section 3.6).

C13

Line 287:  Term “correlation” in Figure 9 is improper according to comments for Figure 2. It can be determined as relationship. Better to use double bar graph type.

Response:

Thank you for the suggestion. We apolagize for the uncorrect description in Figure 9. We have already corrected as the suggestion of the reviewer. We also changed the Figure 9 into double bar graph type.

C14

Line 294:  Number of the Figure should be 10.

Response:

Thank you for the suggestion. We have already corrected on the text (section 3.7).

C15

To check the description of the vertical axis of the Fig.11

Response:

Thank you for the suggestion. The units of C3S is persen (%) because it comes from an equation that uses data on the percentage of chemical composition in the cement sample (Table 2). But, we have changed Figure 11 into Table 3 (section 3.1).

C16

Line 320:  It is written that “All cements have a high Tricalcum Silicate (C3S) content” does not agree with the Fig.11 because the values of the graph of C3S are different, higher and lower. To check values scale on the vertical axis because some of them are negative. Is it possible for C3S  to have values below 0 at Fig.11 ?

Response:

Thank you for the suggestion. Figure 11 has been changed to Table 3 based on data of Table 2 (section 3.1). C3S decreased with the content of Napa Soil. This is because the composition of CaO decreased with the content of Napa soil, while other chemical compositions such as SiO2, Al2O3, and Fe2O3 increased with the content of Napa Soil (Table 2). That’s why C3S possible to have values below 0.

Reviewer 4 Report

General comment
The authors should present a clear objective of the research and write what new elements the research introduces to the world science. It is not known what the mining potential of Napa soil is. Can it be used only locally? Is the potential applicable in the world, in the region?
In the opinion of the reviewer, the authors did not demonstrate the scientific novelty of the article, and therefore I do not recommend it for publication.
Detailed comments
1. The literature review is incomplete. A lot of such research is conducted in the world. There is no need to submit a 1929 article in the context of cement research in the journal like “Materials”. The literature does not include the latest research in this field.
2. I propose to change the organization of the article a little. Chapter 2 needs to be consolidated more, e.g. 2.2. which has one sentence is not good.
3. Chapter 2.2. - how fine are the soil particles? Is the fragmentation homogeneous?
4. There is no need to break it down into 2.4.1, 2.4.2 etc. It is one subsection 2.4
5. Figure 2a is incorrectly described. It is not a correlation. Moreover, in this picture, cPoz8% is misleading. It is not a variable for CNS X%. Likewise, Figures 9 and 10 do not present correlation. Correlation is one number and not a graph.
6. The summary should be expanded.

Author Response

Dear Editor and Reviewers,

Thank you for your feedback on my manuscript entitled “Production of Portland Composite Cement Based on Pozzolan Napa Soil”. After taking into consideration of the comments and suggestions, I have made the necessary changes and correction as required. I wish to resubmit my revised manuscript for your kind consideration for publication. I have also attached herewith my responses to the comments for your information.

Yours faithfully,

(Illyas Md Isa)

Responses to Comments

#Reviewer4

C1

:

The literature review is incomplete. A lot of such research is conducted in the world. There is no need to submit a 1929 article in the context of cement research in the journal like “Materials”. The literature does not include the latest research in this field.

Response:

Thank you for the suggestion. The 1929 literature has been changed by the most recent literature and has been added on the text (References : No.12).

C2

:

I propose to change the organization of the article a little. Chapter 2 needs to be consolidated more, e.g. 2.2. which has one sentence is not good.

Response:

Thank you for the suggestion. We organized already.

C3

:

Chapter 2.2. - how fine are the soil particles? Is the fragmentation homogeneous?

Response:

Thank you for the suggestion. The Napa soil sieved to pass through a 90 µm sieve. This sentence has been added on the text (section 2.2).  Homogeneous.

C4

:

There is no need to break it down into 2.4.1, 2.4.2 etc. It is one subsection 2.4

Response:

We are apologise for that comment. We can’t joint all in one subsection 2.4, because this section have 8 analysis.

Q5

:

Figure 2a is incorrectly described. It is not a correlation. Moreover, in this picture, cPoz8% is misleading. It is not a variable for CNS X%. Likewise, Figures 9 and 10 do not present correlation. Correlation is one number and not a graph.

Response:

We are grateful to you for your detail observation. Figure 2, 4, 5, 6, 7, 8, 9, 10 have been changed into a chart as suggested by reviewer. The ‘correlation’ have been changed into ‘relationship’ on Figure 2, 9, and 10. Meanwhile, CPoz8% is a cement control that contain pozzolan, its works as a comparison of data from cemen containing Napa soil.

C6

:

The summary should be expanded.

Response:

We agree and appreciate the good comments. We have already improved the summary with more relevant facts about this study.

Round 2

Reviewer 4 Report

The article is good organized but the authors should present a clear objective of the research and write what new elements the research introduces to the world science. It is not known what the mining potential of Napa soil is. Can it be used only locally? Is the potential applicable in the world, in the region?
In the opinion of the reviewer, the authors did not demonstrate the scientific novelty of the article, and therefore I do not recommend it for publication.

Author Response

Responses to Comments

C1

:

It is not known what the mining potential of Napa Soil is. Can it be used only locally? Is the potential applicable in the world in the region?

Response:

Thank you for the suggestion. West Sumatera is located in Sumatera Island which is one of the big island in Indonesia. Indonesia is located at the center of the confluence of two young mountains, namely the Mediterranean Circum Mountains and the Pacific Circum Mountains. The territory of Indonesia which is traversed by the Mediterranean Circum Mountains is western Indonesia. Meanwhile, the areas of Indonesia that are traversed by the Pacific Circum Mountains are central and eastern Indonesia. With its geological location, Indonesia is rich in natural resources in the form of mines and minerals, including alumina silicate as Napa soil.

Napa soil produced by nature originating from weathering of fieldspathic rocks and carried out by endogenous forces that do not move from the parent rock or origin. Because this soil does not move, it has a purer nature, in the sense that it is not mixed with other types of soil and materials. Napa soil highly contains SiO2 (63.20 %) and Al2O3 (16.55%). Thus, Napa soil can be used as alternative material as source of silica and it is hoped that it can be used as potential inorganic material as adsorbent, catalyst and additional material in cement industry. In West Sumatra, Napa soils can be found in various areas such as Pesisir Selatan Regency, Tanah Datar Regency, Solok Regency, and Lima Puluh Kota Regency it is estimated to be around 150,000,000 tons (154 Ha). Therefore, Napa soil potential applicable in the world, not locally only.This information had added in the text (section introduction).

Round 3

Reviewer 4 Report

The authors have addressed the  comments.